# Recent Progress in Understanding the Impact of Food Processing and Storage on the Structure–Activity Relationship of Fucoxanthin

**DOI:** 10.3390/foods12173167

**Published:** 2023-08-23

**Authors:** Andrea Gomez-Zavaglia, Lillian Barros, Miguel A. Prieto, Lucía Cassani

**Affiliations:** 1Center for Research and Development in Food Cryotechnology (CIDCA, CCT-CONICET La Plata), La Plata RA1900, Argentina; angoza@qui.uc.pt; 2Centro de Investigação de Montanha (CIMO), Instituto Politécnico de Bragança, Campus de Santa Apolónia, 5300-253 Bragança, Portugal; lillian@ipb.pt; 3Laboratório Associado para a Sustentabilidade e Tecnologia em Regiões de Montanha (SusTEC), Instituto Politécnico de Bragança, Campus de Santa Apolónia, 5300-253 Bragança, Portugal; 4Universidade de Vigo, Nutrition and Bromatology Group, Department of Analytical Chemistry and Food Science, Faculty of Science, E32004 Ourense, Spain; mprieto@uvigo.es

**Keywords:** brown seaweed-derived xanthophyll, chemical structure, isomerization, degradation products, biological activity

## Abstract

Fucoxanthin, a brown algae carotenoid, has attracted great interest because of its numerous biological activities supported by in vitro and in vivo studies. However, its chemical structure is susceptible to alterations when subjected to food processing and storage conditions, such as heat, oxygen, light, and pH changes. Consequently, these conditions lead to the formation of fucoxanthin derivatives, including *cis*-isomers, apo-fucoxanthinone, apo-fucoxanthinal, fucoxanthinol, epoxides, and hydroxy compounds, collectively known as degradation products. Currently, little information is available regarding the stability and functionality of these fucoxanthin derivatives resulting from food processing and storage. Therefore, enhancing the understanding of the biological effect of fucoxanthin derivatives is crucial for optimizing the utilization of fucoxanthin in various applications and ensuring its efficacy in potential health benefits. To this aim, this review describes the main chemical reactions affecting the stability of fucoxanthin during food processing and storage, facilitating the identification of the major fucoxanthin derivatives. Moreover, recent advancements in the structure–activity relationship of fucoxanthin derivatives will be critically assessed, emphasizing their biological activity. Overall, this review provides a critical updated understanding of the effects of technological processes on fucoxanthin stability and activity that can be helpful for stakeholders when designing processes for food products containing fucoxanthin.

## 1. Introduction

Fucoxanthin is a brown-to-olive-green carotenoid pigment, naturally occurring in the chloroplasts of brown algae. It has attracted increasing interest from researchers and industry because of its numerous bioactive activities supported by in vitro and in vivo studies.

From a chemical viewpoint, fucoxanthin is a carotenoid compound with certain peculiarities, including the presence of an unusual allenic conjugated double bond at position 9, a 5,6-monoepoxide, and some oxygen-containing groups (e.g., hydroxyl, epoxy, carbonyl and carboxyl moieties) (Figure 1). *Trans* isomers are those mostly occurring in nature (>88%) [1].

In spite of its attractive properties, fucoxanthin is not widely employed at an industrial level both because of its challenging physicochemical properties (e.g., low solubility, crystallinity) and its instability in food processing conditions. These physicochemical shortcomings have attracted the attention of food and pharmaceutical technologists to develop diverse strategies aiming to overcome them. Nanoencapsulation of fucoxanthin involves the use of nanotechnology to encapsulate or trap fucoxanthin molecules within nanocarriers. This technique has gained interest in recent years due to its potential to improve the bioavailability, stability, and delivery of fucoxanthin. However, there is still a gap in what concerns the fundamentals behind the protection impaired by these systems.

The available literature is generally focused on the extraction, purification, characterization, and determination of the bioactive properties of fucoxanthin, also including attempts to encapsulate it. However, little is known about the products generated through processing, storage, and digestion, which doubtless affect fucoxanthin’s functionality. Even less knowledge is available about the bioactive properties of the degradation products. On the other hand, nanoencapsulation can affect the structure–activity relationship of fucoxanthin by altering its physical and chemical properties. The effects of encapsulation on the biological activity of fucoxanthin may vary depending on the encapsulation method and materials used. Therefore, it is important to carefully evaluate the impact of nanoencapsulation on the biological activity of fucoxanthin in each specific application. Such studies could generate critical knowledge to appropriately define the best processing and storage conditions, enabling the retention of fucoxanthin’s biological properties.

This review critically summarizes the state of the art about the most critical food processing and storage conditions affecting the chemical stability of fucoxanthin. Information about degradation products, including (when available) their potential bioactive properties, is also discussed (Figure 1). Special emphasis was put on the understanding of the effect of technological processes on the stability and activity of fucoxanthin, in order to provide useful information to aid decision-making when designing processes for the development of food products, supplements, or cosmetics containing fucoxanthin.

## 2. Food Processing and Storage Conditions Affecting Fucoxanthin Structure

The use of fucoxanthin as a functional ingredient in the design of novel foods is limited due to fucoxanthin, like most carotenoids, being susceptible to different processing and storage conditions (e.g., temperature, oxygen, light, pH, storage time), and the pro-oxidant components present in the food matrix (e.g., metals, enzymes, among others) [2]. In fact, the propylene group makes fucoxanthin heat-sensitive (positions 13 and 9′ being easily oxidatively cleavable sites), the conjugated polyene long chain makes it photosensitive, and ethylene oxide moiety makes it oxygen-sensitive [3]. Thus, understanding how the mechanisms of food processing and storage affect these functional structures is crucial to design procedures that aim to preserve the molecule and control the formation of undesirable fucoxanthin-derivative compounds [4].

### 2.1. Heat Processing

Heating is the most extensively studied factor influencing fucoxanthin extractability and stability. Some studies have found that temperature has a significant effect on fucoxanthin recovery using pressurized-liquid extraction and microwave-assisted extraction from different macro- and microalgae species [5,6]. Fucoxanthin extractability increases as temperature increases up to a certain value (around 100 °C, depending on species). This effect was attributed to fucoxanthin being coupled to proteins and chlorophyll a, resulting in fucoxanthin-Chl a-protein complexes that can be destroyed by increasing the temperature [7]. Regarding stability, Li et al. [8] found that fucoxanthin degraded as temperature increased from 50 to 200 °C. However, Prabhasankar et al. [9] showed that fucoxanthin remained stable after being subjected to a pasta production process and even during the cooking step, indicating that this pigment can be employed in the food industry. The stability of fucoxanthin can be ascribed to the presence of other antioxidant compounds in the crude extract [10]. Similarly, Nuñez de Gonzalez et al. [11] found that heating fucoxanthin-enriched yogurt at 80 °C for 30 min had no influence on the pigment stability, implying that yogurt appears to be a good matrix to be fortified with fucoxanthin.

Like most carotenoids, fucoxanthin occurs in nature as *trans*-isomers characterized by having low water solubility and a high tendency to crystalize. Its thermodynamic stability is associated with the linear configuration [12]. Heating promotes the isomerization of all *trans*-fucoxanthin to its *cis* counterparts, 13-*cis*, 13′-*cis*, and 9′-*cis* fucoxanthin being the main isomers identified (Figure 2). In this regard, the concentration of 13-*cis* and 13′-*cis* fucoxanthin in canola oil was found to gradually increase as temperature increased to 100 °C, while all *trans*-isomers experienced a drastic concentration decrease ascribed to isomerization (Table 1) [2,13].

Similarly, Honda et al. [1] observed that increasing temperature to 160 °C (higher than usual values) in fucoxanthin supercritical CO_2_ extraction from *Undaria pinnatifida* induced not only isomerization from *trans*-fucoxanthin to its *cis*-counterparts but also enhanced the production of 13-*cis* and 13′-*cis* isomers (Table 1). On the other hand, heating can also promote the degradation of all fucoxanthin isomers (including *trans* and *cis* forms) by cleaving the conjugated double bonds’ polyene chains in the presence of air (oxidation) and light (photodegradation) [2,12]. In this way, 9′-*cis* fucoxanthin isomers were found to degrade at high temperatures as they are more heat-sensitive than 13-*cis* and 13′-*cis* forms (Table 1) [2]. Low temperature (4 °C) is preferred over room temperature to keep fucoxanthin stable during prolonged storage [11,13,14].

### 2.2. Oxygen

The presence of oxygen is inevitable during food processing and storage, and the exposure of fucoxanthin in its free form to oxygen can trigger oxidation reactions. Oxidation can induce cleavage in the conjugated double-bond polyene chain, epoxidation, and hydrolysis of fucoxanthin, depending on the medium conditions (Figure 2) [18,19]. These reactions lead to the formation of new short-chain carbonyl compounds that can be produced on both sides of the fucoxanthin molecule. The generated compounds contain conjugated carbonyl, epoxide, or allenic groups with different chemical properties than their predecessors. Studies dealing with the effect of oxidation on fucoxanthin structure have investigated the exposure of the pigment to air at 25 °C [2] or to different oxidating agents [e.g., ozone, potassium permanganate (KMnO_4_), hypochlorous acid/hypochlorite (HClO/ClO^−^), hydroxyl radicals (OH^−^), and peroxide (H_2_O_2_)] (Table 1) [3,18,19]. From these works, the main products derived from fucoxanthin oxidation can be classified into three groups according to the nature of the oxidized group: fucoxanthinals (aldehydes), fucoxanthinones (ketones), and fucoxanthinol [18,23] (Figure 3). Regarding fucoxanthinals, more than twelve compounds belonging to this group were identified, 15′-apo-fucoxanthinal, 12′-apo-fucoxanthinal and 10′-apo-fucoxanthinal being the most common ones [18,19]. Concerning fucoxanthinones, 9′-apo-fucoxanthinone, 13-apo-fucoxanthinone, and 13′-apo-fucoxanthinone were the main compounds identified [3,18,24]. On the other hand, oxidation can induce the conversion of the fucoxanthin epoxy group into the hemiacetal one, leading to the production of loliolide (Figure 3) [18] Although fucoxanthinol can be produced via fucoxanthin hydrolysis during food processing, it is also the main gastrointestinal metabolite resulting from the consumption of all *trans*-fucoxanthin (Figure 3) [25].

### 2.3. Light Exposure

Like most carotenoids, fucoxanthin is a photosensitive pigment as it can absorb light in the UV and visible regions and reach an excited state more prone to react with the solvent and produce free radicals [18,26]. During the processing and storage of foods, fucoxanthin can be exposed to light, leading to chemical instability. The exposure to light could promote the isomerization of fucoxanthin at an early storage stage and photodegradation in long-term storage (Figure 1, Table 1). For instance, Zhao et al. [2] found that the illumination of fucoxanthin in canola oil at 300 or 2000 lux induced isomerization from 13-*cis* and 13′-*cis* forms to all-*trans* isomers during the first week of storage. In addition, the exposure to light at low intensity (300 lux) significantly promoted the production of 9′-*cis* fucoxanthin from all-*trans* isomers during 15 weeks of storage [2]. At prolonged storage, 13-*cis*, 13′-*cis*, and all trans-fucoxanthin isomers were degraded, while 9′-*cis* forms appeared to be more resistant as they remained stable [2]. Similar findings were obtained for other related marine xanthophyll like astaxanthin. In this sense, Martinez-Delgado et al. [26] found that illumination promoted isomerization of all *cis*-astaxanthin to all-*trans* counterparts during 21 days of storage at 30 °C.

### 2.4. pH

pH medium can also have an impact on fucoxanthin structure. Fucoxanthin has been shown to be unstable under acidic conditions with high degradation rates in all-*trans*- and -*cis* isomers (Figure 1, Table 1) [21]. Fucoxanthin stability decreased during goat milk yogurt fermentation, which was linked to the production of lactic acid, reducing pH and affecting fucoxanthin structure [11]. This low stability was also observed in structurally similar xanthophylls, such as violaxanthin and antheraxanthin, indicating that the epoxy group is very sensitive to low pH [27]. In this regard, the 5,6-epoxide group of fucoxanthin could be converted into 5,8-epoxide via epoxidation and under acidic conditions, leading to the formation of new compounds with different chemical properties. On the contrary, fucoxanthin was more stable in neutral systems during storage [21]. In addition, the formation of 9′-*cis* fucoxanthin was observed during the first half of storage, followed by degradation to initial values until the end of storage [21]. Similarly, violaxanthin and antheraxanthin were stable at neutral conditions since the 5,6-epoxide group did not show any chemical modification [27]. From these findings, high pH values are preferred to preserve fucoxanthin structure, but these values are rarely employed in the food industry.

Overall, the conclusive identification of the primary factors exerting a significant influence on fucoxanthin stability is a challenging task, given the potential occurrence of synergistic and intricate mechanisms, which are further influenced by the specific matrix in which fucoxanthin is dissolved. For example, Zhao et al. [21] demonstrated that the stability of all-*trans* fucoxanthin and its *cis*-isomers against various influencing factors was lower when dissolved in an oil/water emulsion compared to when dissolved solely in oil [2]. This observation was attributed to the presence of a high concentration of available oxygen in the water phase of the emulsion, which triggered oxidation reactions. Conversely, the fucoxanthin extract derived from *S. tenerrimum* demonstrated varying degrees of susceptibility to different environmental conditions, with light being the most influential factor, followed by heat, open air, and room temperature [17].

## 3. Biological Activity of Fucoxanthin Derivative Compounds

Food processing and storage, as previously stated, can have a significant impact on the structure of fucoxanthin. Because the biological activity of fucoxanthin is closely related to its structure (functional groups, conjugated double bonds, stereochemistry), determining the potential bioactivity of fucoxanthin derivatives is of interest. In this regard, the most studied derivatives of fucoxanthin were grouped in Figure 3 according to the primary reaction resulting from different processing and storage conditions. Within each compound grouping, emphasis is directed towards highlighting the functional active group, which holds the potential responsibility for their respective biological activities.

As previously described, isomerization is the most common reaction triggered by different food processing and storage factors. *Trans*-carotenoid isomers are majorly present in nature and are characterized by having a crystalline state and low solubility in certain polar and nonpolar solvents [28]. However, isomerization from *trans*-carotenoids to their *cis* forms induces changes in their physicochemical properties. In general, *cis*-carotenoid isomers have an amorphous state with a lower tendency to crystallize and lower melting points as they are thermodynamically less stable than their *trans* counterparts [29]. In addition, *cis*-carotenoids have shown higher solubility in polar and nonpolar solvents in comparison to their *trans* forms. Although these features were attributed to *cis*-lycopene, β-carotene, and astaxanthin, it could be assumed that *cis*-fucoxanthin isomers follow a similar pattern [30,31].

On the other hand, the effects of isomerization on bioaccessibility, bioavailability, and bioactivity differ among carotenoids. For example, *cis*-lycopene isomers and *cis*-astaxanthin isomers showed higher bioavailability and biological activity than their *trans* forms [32,33]. On the contrary, β-carotene *cis* isomers reported less bioavailability than its *trans*-isomers [33].

The extensive scientific evidence supporting the biological activities associated with all-*trans*-fucoxanthin (Figure 4) has been recently reviewed by Mohibbullah et al. [34] based on in vitro and in vivo studies.

However, the biological effect of fucoxanthin *cis*-isomers and those derivatives resulting from food processing and storage has been scarcely studied and the mechanisms remain unclear (Table 2). In this regard, apo-9′-fucoxanthinone has been gaining attention for its strong anti-inflammatory effect, as this compound can inhibit nitric oxide (NO) and prostaglandin E_2_ production, reduce the expression of inducible nitric oxide synthase (iNOS) and cyclooxygenase-2 (COX-2), and suppress the production of pro-inflammatory cytokines such as tumor necrosis factor-α (TNF-α) and interleukin-6 (IL-6) in in vitro RAW 264.7 cells and in vivo zebrafish models (Table 2). In addition, apo-9′-fucoxanthinone has a multifaceted potential for therapeutic use as it has demonstrated antiproliferative activity against Caco-2 cells, immunomodulation activity, and stimulation of hair growth. Another degradation product with promising anti-inflammatory activity is apo-10′-fucoxanthinal, which has been shown to downregulate the expression of inflammatory mediators by suppressing mitogen-activated protein kinase (MAPK) and nuclear factor-*κ*B (NF-*κ*B) signaling [35]. Thus, the chemical structure of fucoxanthin degradation products play an important role in defining the mechanisms that determine biological activity. Comparing two oxidative products, apo-9′-fucoxanthinone showed higher antiproliferative activity on Caco-2 cells than apo-13- fucoxanthinone, indicating that the allenic structure may be crucial for such activity [3].

The enhancement of antioxidant characteristics in carotenoids has been documented to arise from the presence of functional groups within the terminal rings of these compounds. That is the case of fucoxanthin, in which the allenic bonds [36], and the 5,6-monoepoxide structures are responsible for such properties (Figure 3) [36]. Notably, the allenic bond plays a pivotal role in inhibiting the generation of superoxide and NO [37]. In contrast, the presence of a 4-oxo β-group, as observed in carotenoids such as astaxanthin and canthaxanthin, promotes the generation of NO. Most of the degradation products of fucoxanthin retain such structures and therefore still have antioxidant properties.

The enhancement of fucoxanthin derivatives’ biological activities could also be related to changes in their physicochemical properties, such as higher solubility, which leads to better addition to bile acid micelles and thus higher bioavailability [28].

**Table 2 foods-12-03167-t002:** Changes in biological activity of fucoxanthin derivatives resulting from food processing and storage.

Biological Activity	Fucoxanthin Derivative Compound	Source	Main Findings	Ref.
Anti-inflammatory				
	9′-*cis* fucoxanthin, and a complex including both 13-*cis* and 13′-*cis* fucoxanthin	*Sargassum siliquastrum*	9′-*cis* fucoxanthin inhibited the production of NO, TNF-α, and IL-6 in RAW 264.7 cells. 13-*cis* and 13′-*cis* fucoxanthin showed cytotoxic effect.	[38]
	Apo-9′-fucoxanthinone	*Sargassum muticum*	Suppression of NO and PGE_2_ production, and iNOS and COX-2 expression in RAW 264.7 cells. Reduction of ROS and NO production, and level of iNOS, COX, TNF-α, and IL-1β in LPS-treated zebrafish.	[39]
		*S. muticum*	Suppression of NO and PGE_2_ production.	[40]
		*Sargassum horneri*	Suppression of NO production in LPS-stimulated RAW 264.7 cells.	[41]
	Loliolide	*Sargassum horneri*	Suppression of NO production in LPS-stimulated RAW 264.7 cells.	[41]
		*S. horneri*	Suppression of IL-1β, IL-6, and TNF-α, PGE_2_ COX-2, and iNOS expression in LPS-induced cells.	[42]
	Apo-10′-fucoxanthinal	Prepared from fucoxanthinol	Suppression of MAPK and NF-*κ*B signaling via downregulating the expression of inflammatory mediators in LPS-induced RAW264.7 cells.	[35]
	Fucoxanthinol	*Nitzschia laevis*	Suppression of NO, PGE_2_, and ROS production, and reduction of the expression of iNOS, COX-2, IL-1β, TNF-α, and IL-6, in LPS-induced BV-2 cells.	[25]
Antioxidant				
	Ratio of *trans*- to *cis*-fucoxanthin (100:3:7; 100:3:8, 100:3:10)	*Phaeodactylum tricornutum*	As the concentration of the *cis* isomers increased, the antioxidant activity (DPPH, superoxide anion, reducing power, and hydrogen peroxide) decreased.	[14]
	9′-*cis*, 13-*cis* and 13′-*cis* fucoxanthin isomers	*Laminaria japonica Aresch*	DPPH and superoxide radical scavenging activity: 13-*cis*- and 13′-*cis*-isomers > all *trans*-fucoxanthin > 9′-*cis*-isomer. ABTS scavenging activity: 9′-*cis* isomer > all *trans*-fucoxanthin > 13-*cis* and 13′-*cis* isomers.	[43]
	Fucoxanthinol	Standard	Inhibition of cellular reactive oxygen species accumulation.	[44]
Anticancer				
	Apo-13-fucoxanthinone and apo-9′-fucoxanthinone	*U. pinnatifida*	Apo-9′-fucoxanthinone showed higher cytotoxic effect against Caco-2 cells than apo-13-fucoxanthinone.	[3]
	9′-*cis* fucoxanthin, and a complex including both 13-*cis* and 13′-*cis* fucoxanthin	*Sargassum siliquastrum*	All isomers significantly inhibited human fibrosarcoma (HT1080) cell migration. Reduction of the MMP-2 and MMP-9 activities.	[45]
	Fucoxanthinol	Standard	Reduction of in colorectal cancer cells (HCT116, DLD-1, Caco-2, WiDr, SW620).	[46]
		Enzymatically prepared from fucoxanthin standard	Modulation of gene expression and core signaling pathways in human breast cancer cells (MCF-7 and MDA-MB-231). Induction of apoptosis in MCF-7 and MDA-MD-231 cells.	[47]
Immunomodulation				
	Apo-9′-fucoxanthinone	*S. muticum*	Modulation of immune system by inhibition of IgE serum levels and cutaneous edema. Reduction of IL-4, IFN-γ, and TNF-α production. Decrease of lymph node size in atopic dermatitis mouse.	[48]
Anti-hyaluronidase				
	*Trans*-, 9′-*cis*, and 13′-*cis* fucoxanthin isomers.	*Sargassum vulgare* Keelakarai, *Turbinaria ornata*, *Turbinaria conoides*	Despite *cis*-isomers being able to react with hyaluronidase enzyme through hydrophobic interactions, these forms were less stable than *trans*-fucoxanthin.	[17]
Hair growth stimulation				
	Apo-9′-fucoxanthinone	*S. muticum*	Apo-9′-fucoxanthinone induced the dermal papilla cell growth and reduction of 5α-reductase activity, suggesting its potential use for hair growth.	[49]

NO: nitric oxide; TNF-α: tumor necrosis factor-α; IL-6: interleukin-6; PGE_2_: prostaglandin E_2_; iNOS: inducible nitric oxide synthase; COX-2: cyclooxygenase-2; IL-1β: interleukin-1β; LPS: lipopolysaccharide; IgE: immunoglobulin E; IL-4: interleukin-4; IFN-γ: interferon-γ; IL-12 p40: interleukin 12 p40; CpG: cultured glial cells; MAPK: mitogen-activated protein kinase; NF-*κ*B: nuclear factor-*κ*B; MCF-7 and MDA-MB-231: breast cancer cell lines; DPPH: 1,1-diphenyl-2-picrylhydrazyl radical; ABTS: 2-2′-azinobis-(3-ethylbenzothiazoline-6-sulfonic acid); Caco-2: human colorectal adenocarcinoma cells; MMP-2: matrix metalloproteinases-2; MMP-9: matrix metalloproteinases-9; HL-60: human promyelocytic leukemia cells; HCT116: human colorectal carcinoma cell; DLD-1: colorectal adenocarcinoma cell, WiDr: derivative of colon adenocarcinoma cell line, HT-29; SW620:human Caucasian colon adenocarcinoma.

On the other hand, there is an increasing interest in studying fucoxanthinol as it has shown anti-inflammatory, antioxidant, neuroprotective, and antiproliferative activity against different cancer cell lines (colorectal, breast, etc.) [25,44,50]. Li et al. [25] found that fucoxanthinol extracted from *Nitzschia Laevis* exhibited remarkable anti-neuroinflammatory effects. These authors observed a significant reduction in the production of various pro-inflammatory substances, such as nitric oxide (NO), prostaglandin E-2 (PGE-2), reactive oxygen species (ROS), interleukin-1 (IL-1), interleukin-6 (IL-6), TNF-α, COX-2, and iNOS in BV-2 cells induced with lipopolysaccharide (LPS). Additionally, Li et al. [25] found that fucoxanthinol showed anti-fibrogenic and antioxidant properties by suppressing the expression of pro-fibrogenic genes induced by transforming growth factor β1 (TGFβ1) and reducing the accumulation of ROS in LX-2 cells.

Regarding fucoxanthin isomers, 9′-*cis* and 13′-*cis* fucoxanthin showed higher anti-hyaluronidase activity and instability than *trans* isomers [17]. These authors also concluded that although *cis* isomers are highly active, the biological activity of fucoxanthin is attributed to the *trans* configuration [17]. However, 9′-*cis* fucoxanthin showed higher anti-inflammatory and antioxidant activity (evaluated through ABTS assay) than all *trans*-, 13-*cis*, and 13′-*cis* isomers (Table 2). Heo et al. [38] observed that 9′-cis fucoxanthin exhibited a more pronounced anti-inflammatory effect on the transcriptional activity of NO, TNF-a, and IL-6 within LPS-stimulated RAW 264.7 cells, compared to 13′-*cis* fucoxanthin. Notably, 9′-*cis* fucoxanthin demonstrated the ability to attenuate the expression of iNOS protein and mRNA, concomitant with the reduction of TNF-a and IL-6 production. Conversely, 13′-*cis* fucoxanthin demonstrated a cytotoxic effect. Thus, further in vivo studies are needed to provide comprehensive knowledge regarding the biological activity of fucoxanthin isomers.

Miyashita et al. [51,52] suggested that the allenic moiety is involved in the anti-obesity properties of fucoxanthin. Indeed, carotenoids containing an allene bond and an additional hydroxyl substituent on the side group, such as fucoxanthin and its biologically active form, fucoxanthinol, show suppressive effects on adipocyte differentiation within 3T3-L1 cells. From a biological viewpoint, the anti-obesity effect of fucoxanthin is intricately linked to its influence on the protein and gene expressions of uncoupling protein 1 (UCP1) in white adipose tissue (WAT) [53]. The induction of UCP1 in abdominal WAT by fucoxanthin intake is in part attributed to the upregulation of β3 adrenergic receptor. This receptor is recognized for its involvement in processes such as lipolysis and thermogenesis [53,54], contributing to the oxidative breakdown of fatty acids and heat generation within mitochondria of WAT [52]. Consequently, it can be inferred that the allenic moieties and hydroxyl substituents present in the side chain play a significant role in the regulation of UCP1 expression within white adipose tissue.

Regarding skin protection, fucoxanthin exhibits a suppressive influence on the enzymatic activity of tyrosinase, melanin synthesis within B16 melanoma cells, and mitigates skin pigmentation in UVB-irradiated guinea pigs. Additionally, it also suppresses mRNA expression of various molecules, including COX-2, endothelin receptor A, p75 neurotrophin receptor, prostaglandin E receptor 1, melanocortin 1 receptor, and tyrosinase-related protein 1 [55]. The ability of both fucoxanthin and its degradation byproducts to effectively scavenge free radicals contributes to their capacity to support the inherent defense mechanisms of the skin. However, it is noteworthy that the exploration in this area is still relatively limited, necessitating further comprehensive investigations to elucidate the complete scope of these effects and their practical applications in skincare.

As a whole, it should be mentioned that the available information about the bioactive properties of fucoxanthin and its degradation products is rather descriptive; that is, different assays supporting them have been successfully carried out, but such properties are ascribed to the compound itself, without identifying which of the fucoxanthin functional groups are responsible for them. This underlines the need to set up more precise experiments blocking or inhibiting the different fucoxanthin functional groups in order to identify the role of each of them in the different bioactive properties.

## 4. Conclusions

In spite of the hard work carried out on the sustainable extraction, characterization, and biological properties of fucoxanthin, there still exist important gaps in fundamental knowledge that should be filled for a more accurate development of innovative applications. As mentioned in this review, the low solubility of fucoxanthin represents a challenge both for the development of food and pharmaceutical applications and also for its absorption at the intestinal level. Overcoming this problem requires innovation aiming to enhance its solubility and consequently its bioavailability. This represents a great opportunity to incorporate innovative approaches based on novel technologies, such as micro- and nanotechnology.

Although isomerization reactions are the main degradation paths of fucoxanthin, little is known about the physicochemical and bioactive properties of degradation compounds, which are mainly *cis*-isomers. The amorphous characteristics of other *cis*-carotenoids can be cautiously extrapolated to fucoxanthin. It is well known that amorphous states provide better storage conditions, extending the shelf-life of different food and pharmaceutical products [56]. Therefore, degradation reactions leading to *cis*-fucoxanthin would provide more soluble compounds which could be better incorporated into bile acid micelles and thus be more bioavailable. In spite of that, as the available information about the bioavailability and bioaccessibility of these compounds is still scarce, this technological advantage should be carefully investigated.

Regarding bioactivity, certain *cis*-isomers and other degradation products of fucoxanthin have demonstrated bioactive properties, in particular anti-inflammatory, antioxidant, and antiproliferative ones.

Considering that the main problems to extend the industrial production of fucoxanthin arise from its physicochemical and stability properties, and certain degradation products might have better solubility and bioactive properties, it appears that degradation reactions are not so undesirable as one can *a priori* prejudice. This points out the need for deeper insights on fundamental research to shed light on the mechanisms explaining their physicochemical and biological behavior, and also on in vivo and clinical studies of the degradation products. Such information will be a great help for optimizing the production of degradation products with better physicochemical and bioactive properties. In addition, it will strongly contribute to accurately engineering the processing conditions and stimulate larger industrial applications.

## Figures and Tables

**Figure 1 foods-12-03167-f001:**
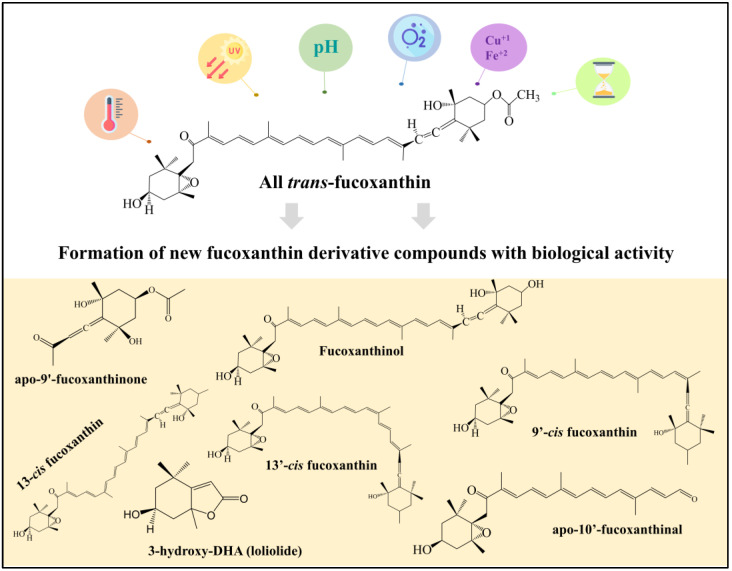
Overview of the structure–function relationship of fucoxanthin and its derivatives.

**Figure 2 foods-12-03167-f002:**
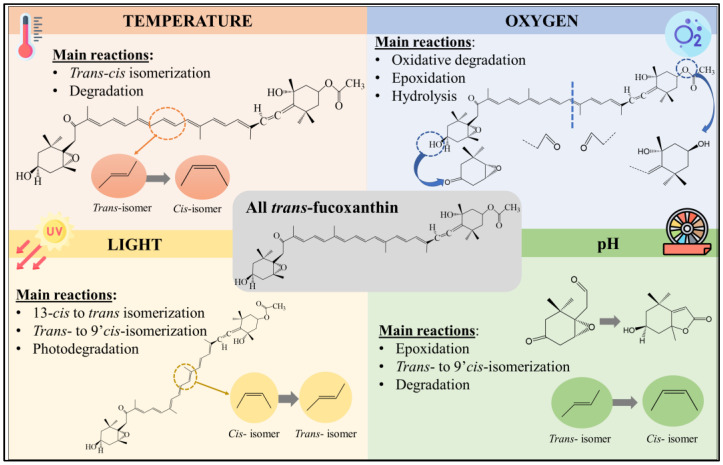
Reactions promoted by the main food processing and storage conditions affecting the fucoxanthin structure.

**Figure 3 foods-12-03167-f003:**
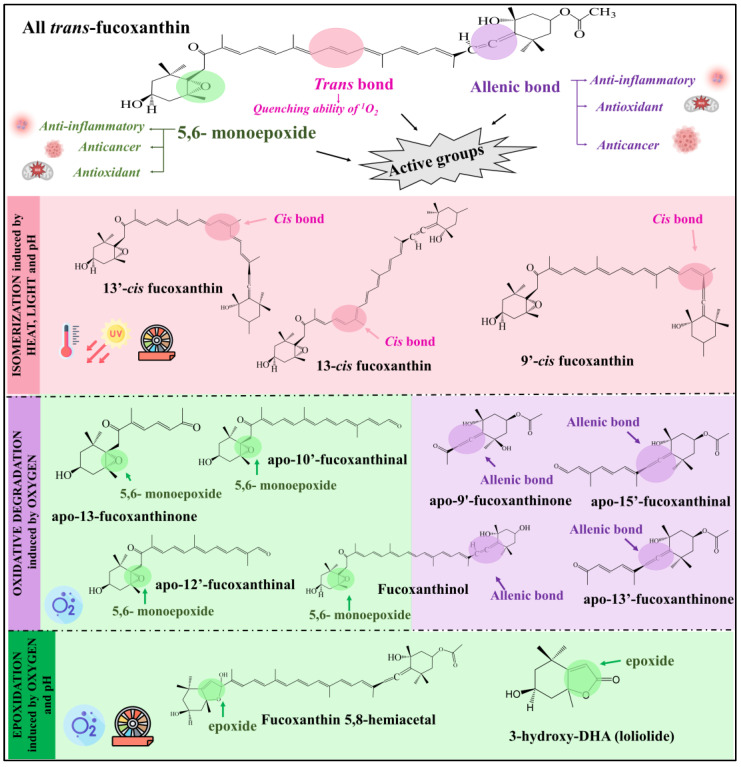
Some derivatives of fucoxanthin systematically organized based on their respective degradation reactions resulting from food processing and storage. Within each grouping of compounds, there exists a highlighted functional active site, which potentially plays an essential role in maintaining the biological activity attributed to fucoxanthin.

**Figure 4 foods-12-03167-f004:**
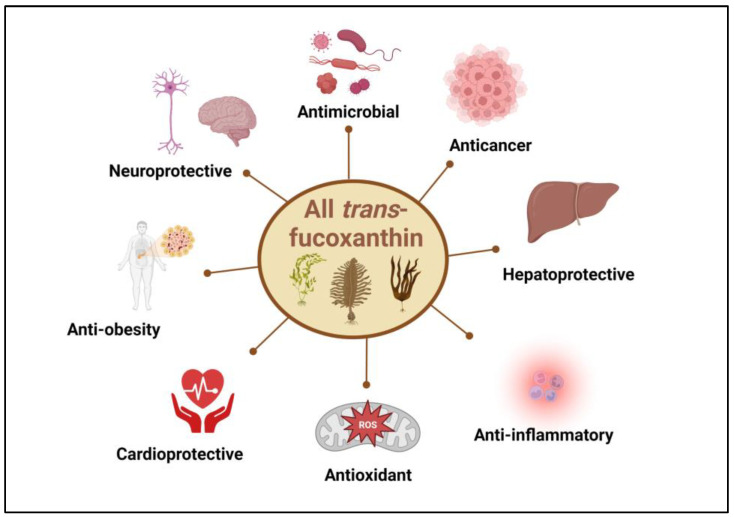
Main biological activities associated with all-trans-fucoxanthin.

**Table 1 foods-12-03167-t001:** Food processing and storage conditions affecting fucoxanthin stability.

Factor	Treatment Conditions	Algae	Main Findings	Ref.
FOOD PROCESSING				
** *Temperature* **				
	Exposure at 25, 37, 50, 80, and 100 °C.	*Phaeodactylum tricornutum*	Fucoxanthin content slightly decreased from 25 to 80 °C. Beyond this value, a significant reduction was observed.	[14]
	Supercritical CO_2_ extraction (40–160 °C, and 20–40 MPa) using ethanol as co-solvent.	*Undaria pinnatifida*	Increasing extraction temperature promoted isomerization from all *trans* fucoxanthin to *cis* forms and recovery of 13-*cis* and 13′-*cis* isomers.	[1]
	Oven drying (40–100 °C).	*Padina australis*	Isomerization from all *trans* fucoxanthin to 13′-*cis*, 13-*cis*, and 9′-*cis* isomers (from 40 to 80 °C). Beyond 80 °C, degradations of all *trans* and *cis* isomers were found.	[13]
	Hot water blanching (HWB) and saltwater blanching (SWB, 0.5–16 min at 60–98.3 °C). Steam blanching (SB, 0.5–16 min at 100.1 °C). Microwave blanching (MWB, 0.5–16 min at 560 W).	*Sargassum fusiforme*	HWB allowed higher fucoxanthin recovery in comparison to other treatments. However, fucoxanthin degraded over time in all treatments applied.	[15]
	Tomato purees supplemented with microalgae and subjected to sterilization.	*Isochrysis*, and *Phaeodactylum*	Fucoxanthin content decreased by 40% due to sterilization	[16]
	−20 °C and room temperature.	*Sargassum tenerrimum*	*Trans*-fucoxanthin isomers remained stable when stored at both temperatures in the dark.	[17]
** *Oxygen* **				
	Exposure to irradiation (10, 20, 40, 60, 80 min) and hydrogen peroxide (20 mg/L).	Standard	Fucoxanthin showed the highest degradation rate. Degradation products were identified as: fucoxanthinals (12′-apo-fucoxanthinal and 15′-apo fucoxanthinal), fucoxanthinones (9′- apo-fucoxanthinone, 13′-apo-fucoxanthinone and 13-apo-fucoxanthinone), fucoxanthinol, and 3-hydroxy-DHA (loliolide).	[18]
	Exposure to potassium permanganate (KMnO_4_) and hypochlorous acid/hypochlorite (HClO/ClO^−^).	*U. pinnatifida*	Fucoxanthin degradation led to the formation of many cleavage compounds such as 3 apo-fucoxanthinones and 11 apo-fucoxanthinals.	[19]
	Ozone oxidation.	*U. pinnatifida*	Fucoxanthin oxidation led to the formation of apo-13-fucoxanthinone and apo-9′fucoxanthinone.	[3]
	Open air in amber color bottle at 30 °C.	*S. tenerrimum*	Approximately 55% of all-*trans* fucoxanthin was retained, while 30% of the fucoxanthin was observed to be oxidized.	[17]
** *Light* **				
	Direct daylight (2500 lux; 90 min) at room temperature	*U. pinnatifida*	Fucoxanthin content was reduced by 90%.	[20]
	Light at 175.77 mol/m^2^/s, room temperature	*S. tenerrimum*	Fucoxanthin underwent oxidation after exposing to light.	[17]
** *pH* **				
	Exposure at pH 2, 4, 6, 7, 8, and 10.	*Phaeodactylum tricornutum*	Fucoxanthin was sensitive to acidic conditions (pH 2–4), but stable in the neutral and alkaline systems (pH 6–10).	[14]
**STORAGE CONDITIONS**				
** *Temperature* **				
	4 and 25 °C for six months.	*Phaeodactylum tricornutum*	Fucoxanthin stored at a low temperature (4 °C) was more stable during long-term storage than at 25 °C.	[14]
	Incubation at 25, 37, and 60 °C in a water bath in the dark.	*C. costata*	Degradation of all *trans*-fucoxanthin and 13-*cis* and 13′-*cis* isomers were observed at every temperature assayed during storage. However, the concentration of 9′-*cis* fucoxanthin was stable regardless of temperature studied during storage.	[21]
** *Oxygen* **				
	Sterilized tomato purees supplemented with microalgae stored for 12 weeks at 37 °C.	*Isochrysis, and Phaeodactylum*	Fucoxanthin was significantly degraded in purees during storage.	[16]
	Incubated in an oven in open air at 25 °C for 30 weeks.	*C. costata*	Fucoxanthin was degraded, resulting in the predominant formation of 9′-*cis* as the major product.	[2]
** *Light* **				
	Exposure at 300 or 2000 lux and incubation for 16 weeks.	*C. costata*	Isomerization from 13-*cis* and 13′-*cis* to all *trans* fucoxanthin was found at both light intensities. Isomerization from all *trans* to 9′-*cis* fucoxanthin was also detected and was more pronounced at 2000 lux. From half of the storage to the end, photodegradation of all *trans* and *cis* isomers was detected.	[2]
	Storage at room temperature with or without light for 30 days.	*Sargassum thunbergii*	A slight reduction of fucoxanthin content was observed during storage in the dark. On the contrary, a drastic decrease in fucoxanthin content was found when stored with light.	[22]
** *pH* **				
	Exposure at pH 1.2, 4.6, and 7.5 and incubation for 120 days.	*C. costata*	All *trans* and *cis*-fucoxanthin isomers were degraded at pH 1.2. However, in neutral conditions, degradation rate of all *trans*, 13-*cis* and 13′-*cis* fucoxanthin was reduced. The formation of 9′-*cis* isomer was observed.	[21]

## Data Availability

The data used to support the findings of this study can be made available by the corresponding author upon request.

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
