# Peer review of "Recent Progress in Understanding the Impact of Food Processing and Storage on the Structure–Activity Relationship of Fucoxanthin"

_foods, 2023, doi:10.3390/foods12173167_

Round 1

Reviewer 1 Report

-The title sounds like a research article. A term indicating that the title is compilation should be used.

-Abstract should be developed. Rather, the importance of fucuxanthi was mentioned.

-There should be flow integrity in the topics. Some similar topics should be combined in the article and comments should be added with the results of the research.

--What are the main aspects that fucoxanthin is affected by in storage, more topics and comments should be added on these. In addition, what are the main factors that are effective in processing. Which of these are the most effective and what is the mechanism of action on the structure of the compound, should be explained.

-References should be checked in text and ref list.

need minor revision

Reviewer 2 Report

I have conducted a thorough review of the manuscript titled "Effect of food processing and storage on fucoxanthin structure-activity relationship," which addresses an important topic for the scientific community, particularly given the increasing interest in the potential health benefits of fucoxanthin. However, I have encountered several significant concerns in the review article . To substantiate my decision, I have provided detailed comments to the authors, urging them to revise and enhance the content of the manuscript.

The primary focus of the review is on the "fucoxanthin structure-activity relationship"; however, I find the content of the manuscript to be unsatisfactory in this regard. I request the authors to clarify where in the manuscript they have discussed how specific "abc" food processing parameters induce "xyz" changes in the fucoxanthin structure, consequently resulting in "rst" activity. The first part of the review predominantly explains how processing conditions influence the variation in fucoxanthin concentration, while the second part focuses on the biological activities of fucoxanthin. Regrettably, Table 2 does not provide information on food processing conditions and their correlation with different types of biological activities of fucoxanthin. Overall, the manuscript's content fails to justify its title, and the lack of sufficient information in the tabular section .

I recommend that the authors incorporate a comprehensive diagram in the manuscript to illustrate clearly how various types of food processing conditions impact the fucoxanthin structure and how such structural variations result in specific bioactivities of fucoxanthin. Addressing these concerns and revising the manuscript accordingly would significantly enhance its quality and relevance.

Not a big issue with English.

Reviewer 3 Report

The manuscript 'Effect of food processing and storage on fucoxanthin structure-activity relationship' is dealing with the interesting subject. Although, there is no full review, only a collection of descriptive data with no in-depth discussion of the findings published in experimental publications. The majority of the research described basic details irrelevant to the researched issue or well-known reported data. In Table 1 could you divide food processing  and storage conditions in two groups?

Reviewer 4 Report

The authors wrote a review concerning fucoxanthin under title "Effect of food processing and storage on fucoxanthin structure activity relationship"

The review covered a potential issue on different processing of fucoxanthin and its structure activity relation.

The authors did a graeeat job in summarizing the different methods of processing in addition to the biological activity of fucoxanthin & drivatives still the review lack the following recommendations:

My recommendations:

1. The review just requires a graphical abstract to summarize the main points in it and it should be attrcative to the reader.

2.The authors also need to add a figure to summarize the main biological activities of fucoxanthins.

3. Figure 1: the stuructures are too small in size

4. In Figure 1: it is not understood the biological activities mentioned for all fucoxanthin derivatives ??? what was the aim of adding those activities ??? This part better to be explained in text and then you can refere to Figure 1

Round 2

Reviewer 1 Report

After correction, this article may be published in foods.

English minor must be checked

Reviewer 2 Report

The manuscript improved significantly. It may be accepted for publication. 

Reviewer 3 Report

Authors have significantly improved the manuscript  'Effect of food processing and storage on fucoxanthin structure-activity relationship' thus I suggest the acceptation in present form.